The association of maternal pre-pregnancy body mass index with macrosomia: a birth cohort study from China

Yan Mingxin 1 2
Zhang Yunbo 2
Zhao Doudou 3
Zhao Yan 4
Liu Danmeng 3
Shan Li 3
Mi Yang 3
Pei Leilei 2 pll_paper@126.com
Qu Pengfei 3 5 xinxi3057@163.com
1 Institute for Hygiene of Ordnance Industry , Xi’an , China
2 Department of Epidemiology and Biostatistics, School of Public Health, Xi’an Jiaotong University Health Science Center , Xi’an, Shaanxi , China
3 Translational Medicine Center Northwest Women’s and Children’s Hospital , Xi’an, Shaanxi , China
4 The First Affiliated Hospital of Xi’an Jiaotong University Dermatology , Xi’an, Shaanxi , China
5 Central Laboratory, Beijing Obstetrics and Gynecology Hospital, Capital Medical University , Beijing , China
Fujioka Kazumichi
Electronic publication date: 2025 Nov 10
Publication date: 2025
Volume: 13
Electronic Location ID: e20332
Received 2025 Apr 28; Accepted 2025 Oct 13
Copyright: © 2025 Yan et al.
Copyright year: 2025
Copyright holder: Yan et al.
License: This is an open access article distributed under the terms of the Creative Commons Attribution License, which permits unrestricted use, distribution, reproduction and adaptation in any medium and for any purpose provided that it is properly attributed. For attribution, the original author(s), title, publication source (PeerJ) and either DOI or URL of the article must be cited.
License URL: https://creativecommons.org/licenses/by/4.0/

Keywords: Pre-pregnancy BMI, Macrosomia, Grade 1 macrosomia, Underweight, Obesity, Birth weight, Overweight

Funding: National Natural Science Foundation of China 82103924 and 72174167 This work was supported by the National Natural Science Foundation of China (grant numbers 82103924 and 72174167). The funders had no role in study design, data collection and analysis, decision to publish, or preparation of the manuscript.

==============================
Objective

To investigate the association between pre-pregnancy body mass index (BMI) and the risk of macrosomia through a preconception-early pregnancy-birth cohort in China.

Methods

Among the 12,254 women initially recruited between July 2018 and December 2021, a total of 11,438 (drop out rate: 6.66%) mother–infant pairs were included in the final analysis after excluding participants with missing data on key variables or lost to follow-up. We collected basic demographic characteristics and lifestyle behavior information of the subjects through questionnaires and practical measurements, and conducted further follow-up for pregnancy outcomes. The study assessed the association between pre-pregnancy BMI-defined categories (underweight, normal weight, overweight, and obesity) and macrosomia using multivariable logistic regression models, adjusting for sociodemographic characteristics, lifestyle behaviors, and maternal clinical factors during pregnancy. A linear trend test was also conducted. Moreover, we utilized restricted cubic spline models with three knots (placed at the 10th, 50th, and 90th percentiles of BMI) and polynomial regression to investigate the non-linear relationship of pre-pregnancy BMI with macrosomia.

Results

A total of 11,438 subjects were included in this study, among whom 645 infants were diagnosed with macrosomia, resulting in a prevalence of 5.64%. The results indicated that, compared with the normal weight group, overweight women had a significantly higher risk of macrosomia (odds ratio (OR) = 1.66, 95% CI [1.35–2.01]), as did obese women (OR = 1.66, 95% CI [1.13–2.45]), while underweight women had a significantly lower risk (OR = 0.55, 95% CI [0.41–0.73]). A similar association pattern between pre-pregnancy BMI and grade 1 macrosomia was observed, consistent with that for overall macrosomia. The use of restricted cubic splines revealed that the prevalence of macrosomia/grade 1 macrosomia increased with rising pre-pregnancy BMI. Furthermore, when we stratified the data by covariates, the nonlinear relationship between pre-pregnancy BMI and macrosomia/grade 1 macrosomia persisted. The results of the polynomial regression showed a gradual increase in fetal birth weight with increasing pre-pregnancy BMI levels.

Conclusions

Pre-pregnancy overweight and obesity were associated with higher risks of macrosomia. Therefore, these findings suggest that promoting healthy weight management before conception may be an effective public health strategy to reduce the risk of macrosomia and improve perinatal outcomes.

Introduction

Fetal macrosomia, defined as a birth weight of ≥4,000 g, is known to commonly prolong the labor process, increase the rate of cesarean section, postpartum hemorrhage and puerperal infections in mothers, and also lead to fetal injuries and asphyxia during delivery (Araujo et al., 2017; Kc, Shakya & Zhang, 2015; Ouzounian, 2016). Studies have also shown long-term effects of macrosomia, including its influence on physical and intellectual development during childhood and adolescence, as well as an increased risk of chronic diseases such as hypertension and diabetes in adulthood (Azadbakht et al., 2014; Ezegwui, Ikeako & Egbuji, 2011; Mu et al., 2012; Oken et al., 2007). Notably, developed countries have seen a rise of 15% to 25% in macrosomia prevalence over the past few decades (Koyanagi et al., 2013; Salihu et al., 2020). Similarly, developing countries like China have also witnessed a rise in macrosomia due to improved living conditions and economic growth, with the prevalence increasing from 6.9% to 7.8% between 2007 and 2017 (Lin et al., 2021; Zeng et al., 2023; Zhao et al., 2019). Given the significant social and personal burden of macrosomia, it is crucial to identify its risk factors to develop primary prevention strategies.

This increase in macrosomia has coincided with changes in maternal pre-pregnancy body mass index (BMI) in modern society. The prevalence of pre-pregnancy obesity among women of childbearing age in the US has reached 22%, with a 69.3% increase over the past 10 years (Chu, Kim & Bish, 2009; Kim et al., 2007). In China, more recent surveys show that as of 2023, the combined prevalence of overweight and obesity among adult women in ten provinces reached 35.0%, with obesity alone at 17.1%, and that rural women aged 20–49 now experience overweight or obesity at a rate of 24.8% (He et al., 2016; Zhang et al., 2024). Some studies have suggested a potential association between elevated maternal pre-pregnancy BMI and excessive fetal birth weight (Feng et al., 2019; Ouzounian et al., 2011; Yu et al., 2013).

For instance, a study from the ABCD Amsterdam cohort demonstrated a linear association between pre-pregnancy BMI and the child’s weight and BMI at 14 months of age. A one-unit increase in pre-pregnancy BMI resulted in an increment of 29 g (95% CI [19–39]) in weight and 0.041 kg/m2 (95% CI [0.030–0.053]) in BMI (Mesman et al., 2009). Similarly, a study from the Rotterdam cohort indicated that pre-pregnancy overweight or obesity resulted in a 1.30-fold and 1.74-fold increased risk of developing large sizes for gestational age (Ay et al., 2009). However, these studies lacked a specific focus on the Chinese population and instead utilized international BMI standards. Even more importantly, their pre-pregnancy height and weight data were based on self-reporting in questionnaires. Another study in a Chinese population showed that women with pre-pregnancy overweight and obesity were associated with a 1.99-fold and 4.05-fold risk of macrosomia, respectively (Feng et al., 2019), while a cohort study in Taiwan, China, showed that a 6- to 46-fold increase in the risk of macrosomia for women with pre-pregnancy overweight and obesity (Liang et al., 2020). Despite being conducted on a Chinese population, their differences were still relatively large. In addition, a meta-analysis highlighted inconsistent relationships between pre-pregnancy underweight and overweight/obesity in offspring, emphasizing the need for further research (Yu et al., 2013). Therefore, focusing on the Chinese population, the effect of pre-pregnancy BMI on macrosomia needs to be further explored in a large data and more rigorous design.

To address these issues, we conducted a prospective birth cohort study in northwest China. This study aimed to investigate the association between maternal pre-pregnancy BMI and the risk of macrosomia. The findings may offer important evidence to guide early maternal weight management and macrosomia prevention efforts.

Methods

Portions of this text were previously published as part of a preprint (Yan et al., 2023).

Study design and data sources

This study was based on data from a large, prospective preconception-early pregnancy-birth cohort at Northwest Women’s and Children’s Hospital of China. Between July 2018 to December 2021, a total of 12,254 women of childbearing age were recruited during their pre-pregnancy check-ups. Data for the cohort were collected via two primary methods. First, trained investigators (including clinical staff and researchers) conducted face-to-face interviews using a standardized, structured questionnaire to gather comprehensive information on sociodemographic characteristics, lifestyle, dietary nutrition, nutrient supplementation, reproductive history, and maternal health. Second, clinical data, including diagnosis of pregnancy outcome and newborn birth weight, were extracted from the hospital’s electronic medical record system. These clinical measurements were recorded by trained medical staff using standardized procedures immediately after delivery. To ensure data integrity, all survey and medical record data underwent a rigorous secondary quality control process by professional auditors.

For the present analysis, a specific analytical sample was derived from the full cohort, as detailed in the participant flow diagram (Fig. 1). Starting from the 12,254 recruited women, we first excluded 624 participants who were lost to follow-up before delivery and 65 women who did not have a live birth outcome due to pregnancy termination or abortion. To ensure the reliability of our statistical models, we then excluded participants with missing data on essential variables. This consisted of 89 participants with missing pre-pregnancy weight or height and 38 with missing data for key covariates (parity, current gestational diabetes mellitus (GDM), and fetal sex). After applying these exclusion criteria, the final sample for this study comprised 11,438 gestational mothers.

Figure 1 Flow diagram for the study cohort selection.

This study was conducted in accordance with the Declaration of Helsinki and was approved by the Medical Ethics Committee of Northwest Women’s and Children’s Hospital (Approval Number: 2018018; Approval Date: March 16, 2018). The study underwent periodic review every 12 months, with no fixed expiration date, and remained ethically approved throughout the entire study period (July 2018 to December 2021). All research participants were fully informed about the study content prior to participation, provided written consent, and signed informed consent forms.

Pre-pregnancy BMI assessment

The pre-pregnancy BMI of the mothers in this study was calculated from the height and weight values measured at the first antenatal visit (≤12 gestational weeks of pregnancy). We carefully measured height (accurate to 0.1 cm) and weight (accurate to 0.01 kg) of the mothers, who wore light clothing but no shoes. BMI was calculated as weight/height2 (kg/m2). All anthropometric measurements were performed by trained health professionals following standardized protocols. In previous studies, it has been observed that pre-pregnancy height of pregnant women through questionnaires tends to be overestimated and weight tends to be underestimated, resulting in underestimation of BMI, which introduces information bias (Mesman et al., 2009; Stewart et al., 1987). In addition, relevant studies have shown that height and weight measured at the first antenatal visit (≤12 gestational weeks of pregnancy) are highly consistent with pre-pregnancy height and weight (Mamun et al., 2011). Therefore, by utilizing measured height and weight in this study, we were able to minimize these biases and improve the accuracy of our findings.

According to Chinese standards, pre-pregnancy BMI of mothers was categorized as underweight (BMI < 18.5 kg/m2), normal weight (18.5 ≤ BMI < 24.00 kg/m2), overweight (24 ≤ BMI < 28.00 kg/m2) and obesity (BMI ≥ 28 kg/m2) (Hua et al., 2018). Meanwhile, based on the World Health Organization (WHO) criteria, underweight (BMI < 18.5 kg/m2), normal weight (18.5 ≤ BMI < 25.00 kg/m2), overweight (25 ≤ BMI < 30.00 kg/m2) and obesity (BMI ≥ 30 kg/m2) were redefined (World Health Organization, 2000).

Birth outcomes

The primary outcome in this study was macrosomia, defined as infants with a birth weight ≥4,000 g. Birth weight data were obtained from the hospital medical record system, where they are routinely recorded by trained obstetric staff immediately after delivery using calibrated electronic scales. Macrosomia was further classified into three grades (Boulet, Salihu & Alexander, 2004), with Grade 1 representing infants weighing between 4,000–4,499 g, Grade 2 between 4,500–4,999 g, and Grade 3 being infants with a birth weight higher than 5,000 g (Boulet et al., 2003).

Covariates

The main covariates in the study included sociodemographic characteristics, lifestyle behaviors and clinical characteristics of mothers during pregnancy, which might be associated with pregnancy outcomes (Li et al., 2021; Sun et al., 2020). Sociodemographic characteristics included fetal sex (male, female), parity (nulliparous, multiparous), maternal age (≤24 years, 25–29 years, 30–34 years, and ≥35 years), maternal education (high school or less, college/university and postgraduate), maternal ethnicity (Han and other), family socioeconomic status (poor, moderate and rich). To measure the household economic level, we used principal component analysis, incorporating variables such as monthly household income, monthly expenditure, housing type, household appliances, and transportation, to construct a family wealth index and divided it into thirds as an indicator for poor, medium, and rich households (Filmer & Pritchett, 2001).

Lifestyle behaviors included alcohol drinking before or during pregnancy (Yes, No), passive smoking before or during pregnancy (Yes, No). Alcohol drinking included a variety of alcoholic beverages (e.g., white wine, beer, red wine, etc.) before or during the whole pregnancy; Passive smoking was defined as inhaling smoke for more than 15 min per day and at least one day per week before or during pregnancy.

Clinical characteristics included cold/fever before or during pregnancy (Yes, No), folic acid supplementation before or during pregnancy (Yes, No), current gestational diabetes mellitus (Yes, No). Folic acid supplementation before or during pregnancy means taking folic acid from the first 3 months of pregnancy to the time of conception. This time window coincides with the critical period of early placental and fetal development. Current GDM is diagnosed in the middle of pregnancy according to Chinese criteria: Fasting plasma glucose (FPG) ≥5.1 mmol/L is abnormal fasting glucose; 1-h postprandial glucose ≥10.0 moml/L is abnormal 1 h glucose; 2-h postprandial glucose ≥8.5 mmol/L is abnormal 2-h glucose. Those with at least one of the above indicators were diagnosed with GDM (Wei & Yang, 2011).

Statistical analyses

In univariate analysis, categorical variables were expressed as frequencies (n) and percentages (%) and compared between groups using the χ2 test or Fisher’s exact test. Quantitative variables were presented as median and interquartile range (IQR) when non-normally distributed.

In multivariate analysis, we initially employed logistic regression models to examine the correlation between pre-pregnancy BMI of mothers and macrosomia, accompanied by a linear trend test. Subsequently, we investigated this relationship in various subgroups, stratified by maternal age, maternal education, family wealth index, parity, current GDM, and fetal sex. Additionally, we explored the association between pre-pregnancy BMI and the three different grades of macrosomia.

To ensure the robustness of the relationship between pre-pregnancy BMI and macrosomia, we conducted three sensitivity analyses. Firstly, we substituted the China BMI criteria with WHO BMI criteria to investigate the association between pre-pregnancy BMI and macrosomia, replicating all the analyses. Secondly, to assess potential non-linear relationships, we employed restricted cubic splines with three knots placed at the 10th, 50th, and 90th percentiles of BMI distribution. This selection follows Harrell’s recommended approach, providing a flexible yet parsimonious model that avoids overfitting and maintains interpretability. Finally, we applied polynomial regression to assess the linear relationship between pre-pregnancy BMI and birth weight. All statistical analyses were performed using SAS version 9.4 and R version 4.2.0, and two-sided P < 0.05 indicated a significant difference.

Results

Baseline characteristics

A total of 11,438 subjects were included in this study, and pregnant women were divided into four groups based on Chinese BMI criteria: underweight (15.69%), normal weight (67.74%), overweight (13.59%) and obesity (2.98%) (Table 1). The subjects were predominantly aged 25–34 years (86.35%), college/university in education (75.88%), Han in ethnicity (98.61%), and moderate in family wealth index (64.74%).

Table 1 Comparison of baseline characteristics between the four pre-pregnancy BMI groups.

Characteristics	N	Under weight	Normal weight	Overweight	Obesity	χ 2	P value	
Patient number	11,438	1,795	7,748	1,554	341			
Maternal age group, years						174.342	<0.001	
≤24	680	166 (9.25)	403 (5.20)	88 (5.66)	23 (6.74)			
25–29	5,571	1,010 (56.27)	3,773 (48.70)	646 (41.57)	142 (41.64)			
30–34	4,210	540 (30.08)	2,895 (37.36)	630 (40.54)	145 (42.52)			
≥35	977	79 (4.40)	677 (8.74)	190 (12.23)	31 (9.09)			
Educational level						87.827	<0.001	
Below high school	1,298	185 (10.31)	815 (10.52)	231 (14.86)	67 (19.65)			
College/university	8,345	1,362 (75.88)	5,598 (72.25)	1,144 (73.62)	241 (70.67)			
Postgraduate	1,795	248 (13.82)	1,335 (17.23)	179 (11.52)	33 (9.68)			
Ethnicity						0.770	0.857	
Han	11,260	1,770 (98.61)	7,623 (98.39)	1,532 (98.58)	335 (98.24)			
Other	178	25 (1.39)	125 (1.61)	22 (1.42)	6 (1.76)			
Family wealth index						64.902	<0.001	
Poor	1,374	215 (11.98)	879 (11.34)	206 (13.26)	74 (21.70)			
Moderate	7,641	1,162 (64.74)	5,154 (66.52)	1,092 (70.27)	215 (63.05)			
Rich	2,449	418 (23.29)	1,715 (22.13)	256 (16.47)	52 (15.25)			
Parity						79.455	<0.001	
Nulliparous	8,299	1,437 (80.06)	5,590 (72.15)	1,042 (67.05)	230 (67.45)			
Multiparous	3,139	358 (19.94)	2,158 (27.85)	512 (32.95)	111 (32.55)			
Fetal sex						1.769	0.622	
Male	5,853	896 (49.92)	3,982 (51.39)	805 (51.80)	170 (49.85)			
Female	5,585	899 (50.08)	3,766 (48.61)	749 (48.20)	171 (50.15)			
Drinking before or during pregnancy						1.908	0.592	
Yes	371	62 (3.45)	245 (3.16)	49(3.15)	15 (4.40)			
No	11,067	1,733 (96.55)	7,503 (96.84)	1,505 (96.85)	326 (95.60)			
Passive smoke before or during pregnancy						2.175	0.537	
Yes	1,771	291 (16.21)	1,173 (15.14)	252 (16.22)	55 (16.13)			
No	9,667	1,504 (83.79)	6,575 (84.86)	1,302 (83.78)	286 (83.87)			
Cold/fever before or during pregnancy						0.729	0.866	
Yes	2,461	374 (20.84)	1,674 (21.61)	341 (21.94)	72 (21.11)			
No	8,977	1,421 (79.16)	6,074 (78.39)	1,213 (78.06)	269 (78.89)			
Folic acid supplementation before or during pregnancy						273.051	<0.001a	
Yes	11,075	1,795 (100.00)	7,385 (95.31)	1,554 (100.00)	341 (100.00)			
No	363	0 (0.00)	363 (4.69)	0 (0.00)	0 (0.00)			
Current GDM						270.862	<0.001	
Yes	2,796	285 (15.88)	1,789 (23.09)	587 (37.77)	135 (39.59)			
No	8,642	1,510 (84.12)	5,959 (76.91)	967 (62.23)	206 (60.41)			
Note:

a Fisher exact test.

The study showed significant differences between different pre-pregnancy BMI groups in age, education, wealth index, folic acid supplementation, parity, and current GDM, but no statistically significant differences in ethnicity, drinking, passive smoke, cold/fever, and fetal sex.

The association of pre-pregnancy BMI and macrosomia

Overall, a total of 645 (5.64%) cases of macrosomia were found in all infants, including 576 cases (89.30%) of grade 1 macrosomia, 50 cases (7.75%) of grade 2 macrosomia, and 19 cases (2.95%) of grade 3 macrosomia. Birth weight significantly differed among different pre-pregnancy BMI groups, with higher rates in the overweight and obesity groups (P < 0.001). Among pre-pregnancy BMI subgroups, including underweight, normal weight, overweight, and obesity groups, stratified according to Chinese criteria, the incidence rates of macrosomia among infants were 3.12%, 5.37%, 9.14%, and 9.09%, respectively (P < 0.001) (Table 2). Notably, the overweight and obesity groups showed the highest prevalence of grade 1 macrosomia (Table 2).

Table 2 Relationship between macrosomia and pre-pregnancy BMI.

Pregnancy outcomes	N	Under weight	Normal weight	Overweight	Obesity	χ 2 /F	P value	
Macrosomia, n (%)						65.855	<0.001	
No	10,793	1,739 (96.88)	7,332 (94.63)	1,412 (90.86)	310 (90.91)			
Yes	645	56 (3.12)	416 (5.37)	142 (9.14)	31 (9.09)			
Grade 1 macrosomia	576	52 (2.90)	373 (4.81)	125 (8.04)	26 (7.26)	69.821	<0.001a	
Grade 2 macrosomia	50	3 (0.17)	28 (0.36)	15 (0.97)	4 (1.17)			
Grade 3 macrosomia	19	1 (0.06)	15 (0.19)	2 (0.13)	1 (0.29)			
Birth weight (g), Median (IQR)	11,438	3,230.00 (3,000.00, 3,500.00)	3,330.00 (3,060.00, 3,600.00)	3,400.00 (3,100.00, 3,700.00)	3,340.00 (3,060.00, 3,670.00)	114.447	<0.001b	
Notes:

a Fisher exact test.

b Kruskal–Wallis test.

IQR, interquartile range.

After adjusting for all covariates using a logistic model, compared to the normal weight group, the underweight group had a significantly lower risk of macrosomia (OR = 0.55, 95% CI [0.41–0.73]), while the overweight group (OR = 1.66, 95% CI [1.35–2.01]) and obesity group (OR = 1.66, 95% CI [1.13–2.45]) had significantly higher risks of macrosomia. The linear trend tests were significant, indicating that the risk for macrosomia increased with the increment of pre-pregnancy BMI (Table 3). Similarly, compared to the normal weight group, the underweight group had a significantly lower prevalence of grade 1 macrosomia (OR = 0.57, 95% CI [0.42–0.76]), while the overweight group (OR = 1.62, 95% CI [1.31–2.01]) and the obesity group (OR = 1.55, 95% CI [1.02–2.35]) had significantly higher prevalences of grade 1 macrosomia. Furthermore, when compared to the normal weight group, the overweight group had a significantly higher prevalence of grade 2 macrosomia (OR = 2.80, 95% CI [1.47–5.32]), while the obesity group also showed an elevated risk (OR = 3.46, 95% CI [1.19–10.10]). The linear trend test indicated a progressive increase in grade 1 and 2 macrosomia with increasing pre-pregnancy BMI (Table S1). Consistently, in different subgroups stratified by baseline covariates, the relationship between pre-pregnancy BMI and macrosomia was directionally consistent, indicating good result stability (Table S2).

Table 3 Association between pre-pregnancy BMI and macrosomia according to logistic regression analysis.

Variable	Model 1	Model 2a	Model 3b	
OR (95% CI), P	Adjusted OR (95% CI), P	Adjusted OR (95% CI), P	
Pre-pregnancy BMI				
Under weight	0.57 [0.43–0.75], <0.001	0.56 [0.42–0.75], <0.001	0.55 [0.41–0.73], <0.001	
Normal weight	1.00	1.00	1.00	
Overweight	1.77 [1.45–2.16], <0.001	1.79 [1.47–2.19], <0.001	1.66 [1.35–2.01], <0.001	
Obesity	1.76 [1.20–2.58], 0.004	1.80 [1.22–2.64], 0.003	1.66 [1.13–2.45], 0.010	
P for trend	<0.001	<0.001	<0.001	
Notes:

a Model 2 used Model 1 and adjusted for maternal age, education level, ethnicity, and family financial situation.

b Adjusted for Model 2 and drinking before or during pregnancy, passive smoke before or during pregnancy, cold/fever before or during pregnancy, folic acid supplementation before or during pregnancy, parity, current GDM, fetal sex.

Sensitivity analyses

According to the BMI criteria proposed by the WHO, compared with the normal weight group, the underweight group had a significantly lower risk of macrosomia (OR = 0.53, 95% CI [0.40–0.71]), while the overweight group (OR = 1.72, 95% CI [1.37–2.16]) and the obesity group (OR = 2.33, 95% CI [1.44–3.78]) had significantly higher risks. The linear trend test results were consistent with the results based on China criteria (P < 0.001) (Table 4). Using the restricted cubic spline model, results suggested that BMI lower than 25 kg/m2 was associated with a decreased risk of macrosomia/grade 1 macrosomia, while BMI higher than 25 kg/m2 was associated with an increased risk of macrosomia or grade 1 macrosomia (Figs. 2A and 2B). This correlation remains stable in different subgroups stratified by covariates (Figs. S1 and S2). Additionally, the results of the polynomial regression showed a gradual increase in fetal birth weight with increasing pre-pregnancy BMI levels (Fig. S3).

Table 4 Effects of pre-pregnancy BMI on macrosomia based on the BMI criteria proposed by the WHO.

Variable	Model 1	Model 2a	Model 3b	
OR (95% CI), P	Adjusted OR (95% CI), P	Adjusted OR (95% CI), P	
Pre-pregnancy BMI				
Under weight	0.55 [0.42–0.73], <0.001	0.55 [0.41–0.73], <0.001	0.53 [0.40–0.71], <0.001	
Normal weight	1.00	1.00	1.00	
Overweight	1.85 [1.48–2.30], <0.001	1.87 [1.50–2.33], <0.001	1.72 [1.37–2.16], <0.001	
Obesity	2.43 [1.51–3.92], <0.001	2.48 [1.54–4.02], <0.001	2.33 [1.44–3.78], 0.001	
P for trend	<0.001	<0.001	<0.001	
Notes:

a Model 2 used Model 1 and adjusted for maternal age, education level, ethnicity, and family financial situation.

b Adjusted for Model 2 and drinking before or during pregnancy, passive smoke before or during pregnancy, cold/fever before or during pregnancy, folic acid supplementation before or during pregnancy, parity, current GDM, fetal sex.

Figure 2 Association of pre-pregnancy BMI with macrosomia (A) and Grade 1 macrosomia (B).

Adjusted for maternal age, education level, ethnicity, family financial situation, drinking before or during pregnancy, passive smoke before or during pregnancy, cold/fever before or during pregnancy, folic acid supplementation before or during pregnancy, parity, current GDM, fetal sex.

Discussion

Principal findings in dialogue with existing literature

According to this mother-infant cohort study in Northwest China, we found a prevalence of 5.64% of macrosomia in all infants. Pre-pregnancy underweight was associated with a decreased risk of macrosomia adjusting for all possible confounders by logistic regression, while pre-pregnancy overweight and obesity were associated with an increased risk of macrosomia. Moreover, we observed that the risk of macrosomia increased with quantitative pre-pregnancy BMI. Through a variety of sensitivity analysis, this relationship still persisted, suggesting that pre-pregnancy BMI is strongly associated with macrosomia.

In our cohort study, women with overweight and obesity had a 1.66-fold increased risk of macrosomia, compared to the normal weight group. A meta-analysis of 16 cohort studies from different countries found that pre-pregnancy obesity was associated with an almost two-fold higher odds of macrosomia (Dai, He & Hu, 2018). Based on findings from a prenatal cohort study, each 1 kg/m2 increase in maternal pre-pregnancy BMI was associated with a 5.21 g increase in neonatal adiposity, a 7.71 g increase in defatted body weight, and a 0.12% rise in body fat percentage (Starling et al., 2015). Previous studies suggested that pre-pregnancy overweight and obesity are important risk factors for pregnancy complications and adverse perinatal outcomes (Dempsey et al., 2005; McDonald et al., 2010). Our study results was consistent with some researches that also focused on Chinese. In a Chinese cohort study that included 20,321 mothers and infants, pre-pregnancy overweight and obesity increased the risk of macrosomia by 1.99-fold and 4.05-fold, respectively (Feng et al., 2019). Similarly, in another Chinese cohort study, pre-pregnancy overweight and obesity increased the risk of macrosomia by 1.92-fold and 2.48-fold, respectively (Sun et al., 2020). A meta-analysis, including 45 studies, showed that maternal pre-pregnancy overweight and obesity increased the risk of macrosomia by 1.67-fold and 3.23-fold, respectively among infants (Yu et al., 2013).

In our study, underweight mothers have a 0.55-fold decreased risk of macrosomia in offspring, compared to mothers with the normal weight group. Past findings on the association between pre-pregnancy underweight and macrosomia are inconclusive. Liu et al. (2016) systematically reviewed 60 related studies and reported a negative association between low pre-pregnancy BMI and macrosomia. In a large cohort study of 105,768 mother-infant pairs, Li et al. (2021) demonstrated a correlation between pre-pregnancy underweight and the occurrence of macrosomia, which persisted after adjusting for covariates. However, a recent cohort study that included 2,210 women found no significant association between pre-pregnancy underweight and macrosomia (Liang et al., 2020). The discrepancy in the results may be due to the small sample size in this study. Our findings are consistent with most current studies suggesting that pre-pregnancy underweight is associated with a decreased risk of macrosomia. However, previous studies have shown that pre-pregnancy underweight increased the risk of small-for-gestational-age (SGA) and low birth weight (LBW) (Yu et al., 2013). Therefore, it may be possible to decrease the risk of macrosomia by regulating weight before pregnancy, but it should be kept within a certain range to prevent an increased risk of other adverse pregnancy outcomes. Further studies should focus on the range of pre-pregnancy weight regulation that decreases the risk of macrosomia without increasing the risk of other adverse pregnancy outcomes.

Potential mechanisms and strengths of the study

Several mechanisms have been proposed to explain the association between pre-pregnancy overweight and obesity and macrosomia. First, pre-pregnancy overweight and obesity may lead to the increased concentrations of glucose, amino acids and free fatty acids in the pregnant woman’s body, thereby increasing the risk of abnormal birth weight in the baby (Hull et al., 2011). Secondly, high pre-pregnancy BMI may lead to an abnormal distribution of adipose tissue, disrupting metabolic and immune functions, and may affect the intrauterine environment during pregnancy, resulting in fetal dysplasia and the development of macrosomia (Alberico et al., 2014). Additionally, studies confirm that adipose tissue is resistant to insulin function, further amplifying the risk of fetal macrosomia (Agha, Agha & Sandall, 2014; Vrachnis et al., 2012).

In contrast, the biological mechanisms underlying the inverse association between underweight and macrosomia are less well understood. One plausible explanation is that maternal underweight may reflect reduced energy and nutrient reserves, which could limit placental function and fetal growth potential. Inadequate maternal fat stores might also impair the intrauterine nutrient supply, thus reducing the likelihood of excessive fetal weight.

Furthermore, we adopted restricted cubic splines to explore the association between pre-pregnancy BMI and macrosomia. The results showed that as pre-pregnancy BMI increased, the risk of macrosomia among infants progressively ascended. The results of the study remained stable in the subgroups stratified by covariates. Moreover, polynomial regression was further used to test the linear relationship between pre-pregnancy BMI and birth weight of infants. Maternal pre-pregnancy BMI was found to be linearly related to neonate birth weight. These results of restricted cubic splines and polynomial regression confirmed the effects of maternal pre-pregnancy body mass index on neonate macrosomia, and were consistent with the conclusion of logistic regression. From different perspectives, it was clear that the high correlation between pre-pregnancy BMI and macrosomia was confirmed separately.

The present study has the largest advantage of its birth cohort design. Data collection through follow-up interviews in conjunction with a hospital medical record system had a low rate of missing visits and provided strong evidence of causal association. Moreover, we conducted a comprehensive analysis using the Chinese and international standards of BMI respectively. In addition, we utilized different statistical models, including logistic regression, restricted cubic spline, and polynomial regression, to explore the relationship between the categorical and continuous BMI with macrosomia.

Limitations and future directions

There are several limitations in our study that warrant discussion. First, we did not measure the correlation between gestational weight gain and macrosomia in pregnant women. Previous studies have indicated that pre-pregnancy BMI, rather than gestational weight gain, is more closely correlated with neonatal birth weight (Abdel Moety, Gaafar & Ahmed, 2013). Consequently, pre-pregnancy BMI has been proposed as an independent predictor of birth weight (Moore et al., 2004). Second, although we used height and weight data obtained at the first antenatal visit (≤12 gestational weeks) to minimize recall bias, these may still differ slightly from true pre-pregnancy values. Third, some subgroup analyses were limited by small sample sizes, restricting our ability to detect significant associations in certain strata. Additionally, we lacked detailed data on maternal dietary intake, physical activity, pregnancy-related symptoms (e.g., morning sickness), and antenatal care frequency, all of which may affect maternal BMI, gestational weight gain, and fetal growth. These unmeasured factors may have introduced residual confounding. Nevertheless, we adjusted for a wide range of established covariates and performed sensitivity analyses, which consistently supported our conclusions and strengthen the rigor of this study. From a public health perspective, our findings highlight the importance of achieving an optimal BMI before conception, in line with current maternal health guidelines, and underscore the need to integrate weight management strategies into preconception care programs.

Conclusions

In conclusion, our study indicates that pre-pregnancy overweight and obesity are risk factors for macrosomia, while pre-pregnancy underweight is also associated with macrosomia. Moreover, the results confirm a significant linear trend in the relationship between the continuous pre-pregnancy BMI and birth weight. These findings suggest that women may be able to potentially decrease the risk of macrosomia by managing their weight before conception.

Supplemental Information

Supplemental Information 1 Original data used in this article.

Supplemental Information 2 Supplementary files for the manuscript.

We give sincere thanks to Northwest Women’s and Children’s Hospital of China for their efforts in diagnosing macrosomia. Particularly, we are grateful to all participants and to the staff for their striving to collect data.

Additional Information and Declarations

Competing Interests

The authors declare that they have no competing interests.

Author Contributions

Mingxin Yan conceived and designed the experiments, analyzed the data, prepared figures and/or tables, authored or reviewed drafts of the article, and approved the final draft.

Yunbo Zhang conceived and designed the experiments, prepared figures and/or tables, and approved the final draft.

Doudou Zhao analyzed the data, prepared figures and/or tables, and approved the final draft.

Yan Zhao performed the experiments, authored or reviewed drafts of the article, and approved the final draft.

Danmeng Liu performed the experiments, authored or reviewed drafts of the article, and approved the final draft.

Li Shan performed the experiments, authored or reviewed drafts of the article, and approved the final draft.

Yang Mi performed the experiments, authored or reviewed drafts of the article, and approved the final draft.

Leilei Pei conceived and designed the experiments, analyzed the data, prepared figures and/or tables, and approved the final draft.

Pengfei Qu conceived and designed the experiments, analyzed the data, prepared figures and/or tables, and approved the final draft.

Human Ethics

The following information was supplied relating to ethical approvals (i.e., approving body and any reference numbers):

The Medical Ethics Committee of Northwest Women’s and Children’s Hospital (Approval Number: 2018018).

Data Availability

The following information was supplied regarding data availability:

The raw data is available in the Supplemental File.

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
