# Peer review of "The association of maternal pre-pregnancy body mass index with macrosomia: a birth cohort study from China"

_PeerJ, doi:10.7717/peerj.20332_

## Round 0.1 · original submission · Major Revisions

· Academic Editor

Major Revisions

We request that the authors carefully address and revise the manuscript in response to several issues pointed out by the reviewers.

Reviewer 1 ·

Basic reporting

Overall, the scientific writing of this manuscript presents well-presented data. However, some areas of the manuscript need to be improved.

Experimental design

Need to be explained in detail

Validity of the findings

The discussion and the reference date need to be updated.

Annotated reviews are not available for download in order to protect the identity of reviewers who chose to remain anonymous.

·

Basic reporting

This article is interesting because it focuses on the pre-pregnancy BMI and macrosomia. Macrosomia is related to obesity and NCDs in offspring. The study has a huge sample size. The knowledge gap is quite clear and strong evidence. Overall, this study is valuable and well-designed.

Experimental design

The methods need to add the rationale of settings, calculate the sample size and effect size of the study, rationale of statistics of cubic splines with three knots. Please add the dropout rate that might be used for the future study. Please explain and add the evidence to support why choose the folic acid supplement before and first 3 months of pregnancy and macrosomia.

Validity of the findings

The findings are interesting and respond to the aim of the study. It might be better to add the discussion of dietary, exercise, and physical activity, complications, and symptoms during pregnancy, such as morning sickness, and frequency of antenatal care visits, because it's associated with pre-pregnancy BMI, gestational weight gain, and baby birth weight.

Additional comments

Please add the date of Ethical approval and the period of it.

---

## Round 0.2 · Minor Revisions

· Academic Editor

Minor Revisions

The sample size and sample selection are still not not clear. Please explain it again.

Reviewer 1 ·

Basic reporting

Authors have done the improvements and adjusted to suggestions and comments from reviewers

Experimental design

Authors have done the improvements and adjusted to suggestions and comments from reviewers

Validity of the findings

Authors have done the improvements and adjusted to suggestions and comments from reviewers

Additional comments

Authors have done the improvements and adjusted to suggestions and comments from reviewers

·

Basic reporting

The article is interesting and well response to the recommendations from the reviewers. It presents a relevant and timely study with clear objectives and a logical structure.

Experimental design

It is well deign but the sample size and sample selection might not clear. Please explain it again.

Validity of the findings

It's clear and well response.

Additional comments

The findings in this study supported the pre-pregnancy BMI and obesity are associate with macrosomia.It is important for antenatal care.

---

## Round 0.3 · Minor Revisions

· Academic Editor

Minor Revisions

Please explain about the confounding factors and how to monitor, including providing the rigour of the analysis and recommendations of this findings that link to the policy and guideline in the future. It will help the reader understand this study.

·

Basic reporting

The overview of article is quite clear about the gap of the study, objectives, methods, and results. The author response to the suggestions clearly. Please, explain in details of the confounding factors and how to monitoring including provide the rigour of the analysis and recommendation of this findings that link to the policy and guideline in the future.

Experimental design

The methodology and design of this study is quite clear and concise. Overall, the study is well-structure that relevant with background and the findings. The author response to the suggestions about the sample size, inclusion-exclusion criteria, data collection, data analysis but the confounding factors in this study might not clear. Please reduce repetition about study design and ethical issues. In addition, please explain in details about the rationale for exclusion criteria. Overall is quite well design and methodology.

Validity of the findings

The findings is based on the aims of the study but please reduce repetition and explain more about the clinical significance for observed increases in macrosomia risk in this study and in discussion. Overall the findngs are valid and well reprt based on the objectives of the study..

Additional comments

Please explain about the confounding factors and how to monitoring including provide the rigour of the analysis and recommendation of this findings that link to the policy and guideline in the future. It' good for the reader to well understanding about this study. Overall, this study is great and meet the standards for publication.

---

## Round 0.4 · Minor Revisions

· Academic Editor

Minor Revisions

Please revise according to the following message. Overall, the report of findings are well, concise, and easy reading but the regression reporting of the findings should be considered again. Some sentences might not be clear “1.66-fold increased risk” since “fold” and “OR” are redundant.

“Overweight (OR = 1.66, 95% CI 1.35–2.01) and obese women (OR = 1.66, 95% CI 1.13–2.45) had a significantly higher risk of macrosomia compared with normal-weight women.”

“Underweight women had a significantly lower risk (OR = 0.55, 95% CI 0.41–0.73).”

·

Basic reporting

From the revised version and response letter are quite clear about the suggestions from all reviewer. The abstract is quite clear and well design. The gap of the study quite clearer. The methods quite clear especially samples selection and quality control.

Experimental design

In my opinion, the design and methods are quite clear but the data analysis should be clear about the statistics analysis such as association or correlation.

Validity of the findings

Overall, the report of findings are well, concise, and easy reading but the regression reporting of the findings should be considered again. Some sentences might not clear “1.66-fold increased risk” since “fold” and “OR” are redundant.

“Overweight (OR = 1.66, 95% CI 1.35–2.01) and obese women (OR = 1.66, 95% CI 1.13–2.45) had a significantly higher risk of macrosomia compared with normal-weight women.”

“Underweight women had a significantly lower risk (OR = 0.55, 95% CI 0.41–0.73).”

---

## Round 0.5 · accepted · Accept

· Academic Editor

Accept

Thank you for your thorough revision.